# Toward a Critical Toponymy Framework for Named Entity Recognition: A Case Study of Airbnb in New York City

**Mikael Brunila**[1,2]    **Jack LaViolette**[3]    **Sky CH-Wang**[4]
**Priyanka Verma**[1]    **Clara Féré**[1]    **Grant Mckenzie**[1]
[1]Platial Analysis Lab, Department of Geography, McGill University
[2]Urban Planning and Governance Lab, School of Urban Planning, McGill University
[3]Department of Sociology, Columbia University
[4]Department of Computer Science, Columbia University
`mikael.brunila@gmail.com`

## Abstract

Critical toponymy examines the dynamics of power, capital, and resistance through place names and the sites to which they refer. Studies here have traditionally focused on the semantic content of toponyms and the top-down institutional processes that produce them. However, they have generally ignored the ways in which toponyms are used by ordinary people in everyday discourse, as well as the other strategies of geospatial description that accompany and contextualize toponymic reference. Here, we develop computational methods to measure how cultural and economic capital shape the ways in which people refer to places, through a novel annotated dataset of 47,440 New York City Airbnb listings from the 2010s. Building on this dataset, we introduce a new named entity recognition (NER) model able to identify important discourse categories integral to the characterization of place. Our findings point toward new directions for critical toponymy and to a range of previously understudied linguistic signals relevant to research on neighborhood status, housing and tourism markets, and gentrification.

## 1 Introduction

Places are not only bounded regions and the material forms they contain, but are vested with and shaped by symbolic associations (e.g., Bell, 1997; Gieryn, 2000; Tuan, 1977). The names we use for places—*toponyms*—play a key role in the "production of space" (Lefebvre, 1991), stabilizing the social reality of both place associations and place boundaries in ways that typically reflect power dynamics. The fact that New York City real estate developers have tried to rebrand parts of Harlem as "SoHa" (Davidson and Fagundes, 2019), for example, suggests that the area's previous names were seen as undesirable to the tenants they hope to attract. *Critical toponymy* refers to the field of research that takes as its focus this relationship between places, their names, and the practices and systems of power that link the two (Rose-Redwood et al., 2010).

We apply the critical toponymic perspective to a large dataset of annotated Airbnb listings in New York City, paying particular attention to the socio-spatial circulation of neighborhood names. Airbnb hosts need to communicate aspects of the location of their listings to potential renters. Nearly all hosts rhetorically situate their units in a set of spatial identities and relations, but their ways of doing so are diverse. Some use conventional neighborhood names (perhaps including nearby neighborhoods as well), others describe proximity to nearby landmarks, while still others simply describe accessibility to types of institutions and businesses such as hospitals, police stations, and restaurants. Spatial variation in these linguistic strategies of emplacement is the main object of our analysis. In other words, we ask: What can we learn about urban dynamics from the ways in which residents of different neighborhoods describe their property locations to prospective renters?

Whereas conventional toponymic analysis is usually limited to the semantic content of place names, we expand our focus to include a wide range of linguistic features that reflect spatial relationships (e.g., expressions of proximity, embeddedness, and connectivity)—alongside formal toponyms for neighborhoods, streets, landmarks, businesses, and so on. To extract references to place and spatial relations from unstructured listing descriptions, we train a custom named entity recognition (NER) model on a novel, hand-annotated dataset to identify a range of toponyms and spatial relationships. We analyze these linguistic features alongside a range of sociodemographic variables measured at the neighborhood level, demonstrating multiple associations between toponymic practices and neighborhood status.

In doing so, we offer a number of contributions. First, we expand the methodological and

conceptual scope of critical toponymy. Nearly all toponymic studies have focused on the centralized naming practices of elites such as mayors or large real estate developers, a "top-down" approach (Bigon, 2020, 3). By contrast, our dataset reflects the "bottom-up" toponymic practices of a larger and more diverse set of social actors at a scale impossible without modern computational techniques. In addition, our NER model allows us to expand the range of place-descriptive resources from the semantics of toponyms in isolation to more subtle, variegated, and relational linguistic strategies, while still retaining an orienting emphasis on neighborhood names, boundaries, and their relationship to neighborhood status and change. Thus, we are able to investigate not only what urban areas are called, but also in which socio-spatial contexts they are invoked by name in the first place.

Secondly, we contribute to the growing literature on Airbnb and housing dynamics. Others have shown the ways in which Airbnb and short-term rentals (STRs) more broadly accelerate gentrification, widen the "rent gap" (Smith, 1987), and remove housing from long-term rental markets (Barron et al., 2020; Ayouba et al., 2020; Horn and Merante, 2017; Wachsmuth and Weisler, 2018). Nevertheless, little attention has been paid to the linguistic strategies that mediate economic transactions between hosts and renters. By uncovering linguistic signals related to gentrification and neighborhood status, we offer findings that can be extended to other residential contexts as well — in particular where new avenues for "technologically and culturally driven" gentrification (Wachsmuth and Weisler, 2018) are being opened up by the growing sector of "platform real estate" (Fields and Rogers, 2021).

Finally, we offer two methodological contributions. First, we introduce a new schema for geospatial NER labeling and a corresponding human-annotated dataset with accompanying models. While this data is particular to New York City, our model is able to generalize beyond the annotated entities in the training data (see Appendix B). Second, we propose a set of new lightweight but accurate methods for toponymy resolution and geospatial dependency parsing. To anticipate our results, our models considerably outperform off-the-shelf NER models on our data and task (see Table 1). This suggests that NER applications to social science at the local level will benefit from, if

not require, specialized models such as ours. Our training data and models are publicly available on Github.[1]

## 2 Prior work

### 2.1 Critical toponymy studies and Airbnb

Whereas early studies of toponymy were oriented toward the enumeration, etymology, and taxonomy of place names—with early practitioners likening the toponymist to a "botanical collector" (Wright, 1929, 140)—the "critical turn" (Rose-Redwood et al., 2010; Medway and Warnaby, 2014) since the 1980s has shifted attention toward place naming practices and their relation to social and political life: ethnic tensions, regime changes, collective memory, commercialization, and so on. In imperial contexts, for example, colonizers often rename(d) territories, cities, and streets to reflect their own ethnolinguistic background and political ideals (Carter, 2013; Wanjiru-Mwita and Giraut, 2020), while the re-imposition of indigenous toponyms is an early form of action by many post-liberation groups (Mamvura et al., 2018; Njoh, 2017; Wanjiru and Matsubara, 2017). In specifically urban contexts, critical toponymy has examined street, neighborhood, and landmark names as they relate to politics and diplomacy (Rusu, 2019; Sysiö et al., 2023), to the corporatization of public spaces (Light and Young, 2015) and to shifting neighborhood status hierarchies and gentrification (Masuda and Bookman, 2018; Madden, 2018).

What nearly all of these studies have in common is a focus on the ways in which powerfully situated actors make decisions about "official" place names: mayors renaming streets to honor local heroes, corporations renaming sports stadiums, or real estate developers rebranding low-status neighborhoods in their efforts to attract residents to new apartment compounds. While such actors exert tremendous influence on the toponymic landscape, toponymic adoption by the urban population is neither guaranteed (e.g., Hui, 2019) nor well studied (Light and Young, 2017).

A major contribution here, therefore, is to examine toponymic reference among thousands of Airbnb hosts and customers whose ways of inscribing urban space do not necessarily conform to those of developers or city planners. While GIS scholars have studied Airbnb at length—for example, its patterns of expansion (Gutiérrez et al., 2017) and role

---

[1] https://github.com/maybemkl/airbnb-place-ner

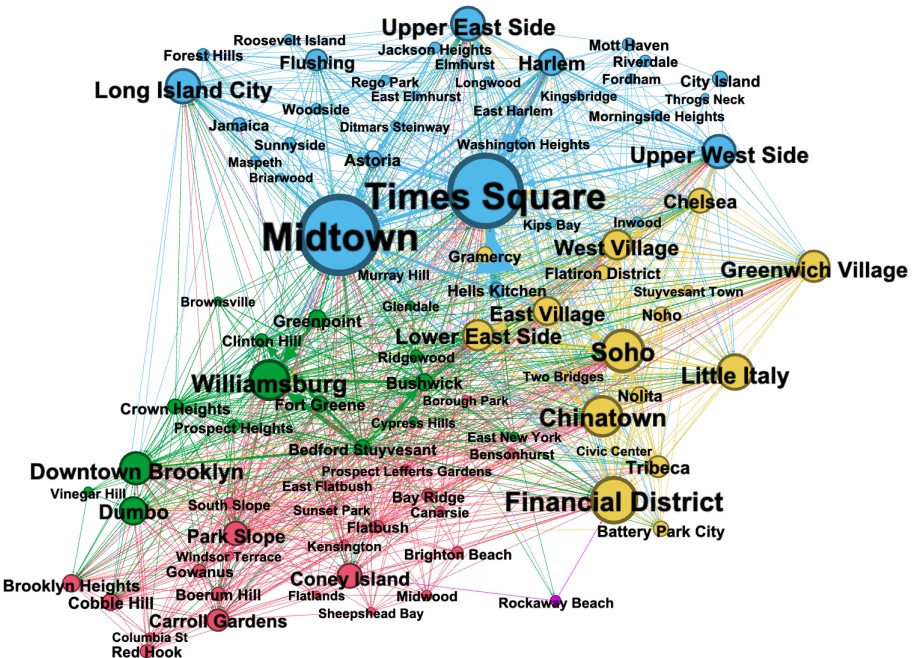

Figure 1: The network of cross-neighborhood mentions in Airbnb listings reflects various geographic, cultural an economic relationships. Here, a weighted directed edge exists between neighborhoods A and B equal to the number of listings located in A that mention B in its description. Colors represent modularity classes. Asymmetries in edge weights tend to reflect prestige and desirability, whereas modularity classes tend to reflect geographic relations. Image filtered to the giant component and nodes with degree of at least 10.

in gentrification (Wachsmuth and Weisler, 2018)—nobody has of yet combined geospatial analysis of Airbnbs with toponymic analysis. In so doing, we heed continued calls to situate embedding-based approaches to language analysis alongside or within other data structures reflecting forms of, for example, spatiotemporal and socio-interactional variation (Bender and Koller, 2020; Brunila and LaViolette, 2022). While NLP methods have previously been used to study Airbnb, we are unaware of previous work that extracts spatial entities and their relations from texts on the platform.

Our approach contributes to a greater understanding of how factors such as gentrification mediate the extent to which cultural signifiers are adopted on the ground. Neighborhood names are particularly suited to this goal, as their boundaries are more ambiguous and fast-changing than other sorts of toponyms. Further, Airbnb data is particularly apt for the critical toponymy perspective, given its simultaneous embeddedness in economic interactions and geospatial structure.

## 2.2 Topynymy and NLP

Beyond the theoretical considerations of critical toponymy, the tasks of extracting place names from unstructured text and determining their geospatial referents—a process called "geo-parsing" (Jones and Purves, 2008, 220)—present numerous practical challenges. The case of New York neighborhoods illustrates this well. For starters, there is no official set of neighborhood names and boundaries in New York. While the NYC Department of Planning offers its own map, it cautions that "neighborhood names are not officially designated,"[2] while a dataset maintained by the nonprofit BetaNYC notes that neighborhood "boundaries may overlap, some neighborhoods may function as a micro-neighborhood within another neighborhood, or a larger district which can be made up of multiple neighborhoods."[3]

Another complication comes from the fact that even if there were a ground-truth dataset of neighborhood names and boundaries, they still need to be mapped to observed tokens in the Airbnb data. Neighborhood names in general, and not least in New York, are often informal and vernacular, including truncations and abbreviations (such as *FiDi* for the Financial District), as well as multilingual

[2]https://www.nyc.gov/site/planning/data-maps/city-neighborhoods.page
[3]https://data.beta.nyc/dataset/pediacities-nyc-neighborhoods

| Model | F1 on Toponyms |
|---|---|
| DistilRoBERTa-CRF | **0.9256** |
| spaCy (LOC+GPE+FAC+ORG) | 0.6175 |
| Stanza (LOC+GPE+FAC+ORG) | 0.5107 |

Table 1: The DistilRoBERTa-CRF model fine-tuned on geospatial NER tags outperforms off-the-shelf models at identifying locations in our test set. To align label schemes, all tags from our model identified as toponyms (tags starting with TN) were recoded to LOC. For the large spaCy model (en_core_web_lg) and the Stanford Stanza model, all tags identified as LOC, GPE, FAC, and ORG were recoded as LOC.

names (such as *El Barrio* for East Harlem) (Hu et al., 2019). In relatively informal written corpora such as Airbnb data, neighborhoods are frequently misspelled or abbreviated in non-standard ways (such as *wb* for Williamsburg).

Thus, from an engineering perspective, toponymic analysis requires two steps: toponymy *detection* and toponymy *resolution* (Wang et al., 2020; Jones and Purves, 2008). First, toponyms have to be detected among the set of words comprising a text, a variant of the general task of named entity recognition. For specialized contexts such as ours, off-the-shelf NER models frequently fail to identify tokens of interest for the reasons mentioned above (again, see Table 1). Secondly, ambiguities introduced by the fact that different names can refer to the same place, and that the same name can refer to different places, are not necessarily identifiable from textual context alone and must be resolved. There is no gold-standard approach to toponymy resolution. Here, we synthesize multiple approaches as described in Section 4.4.

## 3 Data

### 3.1 Airbnb data & neighborhood shapefiles

Our primary dataset contains the 47,440 Airbnb listings that were active in New York City in August 2019, acquired from the nonprofit Inside Airbnb.[4] Each listing is associated with its coordinates (with jitter drawn from a skewed normal distribution with a mean of roughly 200m added), as well as several other variables not relevant here.

As mentioned, numerous neighborhood shapefiles exist for New York. Here we opt for the 264-neighborhood "NYC Neighborhoods" dataset[5]

---

[4]http://insideairbnb.com/
[5]https://data.beta.nyc/dataset/
pediacities-nyc-neighborhoods

maintained by the data science non-profit BetaNYC due to its high granularity. For brevity, we refer to these as "canonical neighborhoods," though we recognize that other neighborhood boundaries exist.

### 3.2 Gentrification index

We adopt the "small area index of gentrification" dataset published as part of Johnson et al. (2022). The authors use changes in sociodemographic variables associated with gentrification from 2000 to 2016 measured at the Census tract level as the basis of their index, which is derived via PCA and Bayesian spatial smoothing. We average these tract-level measures to the neighborhood level, as the obfuscation added to Airbnb coordinates in the form of jitter precludes tract-level analysis.

## 4 Methods

### 4.1 Annotation

To train our model, we developed a 21-category entity taxonomy and hand-labeled these categories across roughly 2,700 listings and reviews. 16 of these categories are toponymic, i.e., apply only when a place is being referenced by name. These include categories for entities such as neighborhoods, streets, transit stations, parks, and businesses. Four of the remaining five categories refer to non-named, generic references to types of institutions whose presence nevertheless meaningfully characterizes urban space. The final category refers not to places or things *per se*, but to spatio-temporal relations: expressions of proximity, distance, and adjacency. By including these final five non-toponymic categories in our model, we expand the range of descriptive strategies people use to construct linguistic cartographies that our model is able to address. Appendix D defines all 21 categories and provides examples.

The schema was developed with initial reference to a set of categories suggested by (Cadorel et al., 2021), and was iteratively expanded and modified from their original 4 labels in Cadorel et al. to 14 labels in some of our preliminary work (Brunila et al., 2023), and finally to the 21 labels used here. All annotation was performed using Prodigy[6] following an initial training session where annotators collaboratively annotated a randomly chosen set of samples. This first round of annotation identified points of ambiguity and disagreement. The

---

[6]https://prodi.gy/

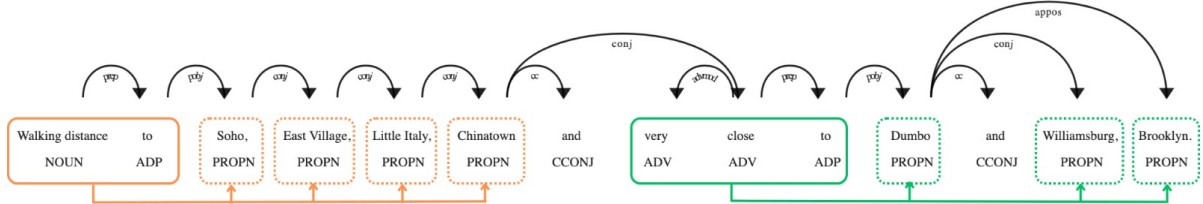

Figure 2: Visualization of the dependency parser. Sentences are searched for dependencies between tokens, merging tokens belonging to the same NER label. The resulting directed graph is filtered for instances where spatio-temporal entities refer to toponyms.

last round of annotation, on which the final model was trained, involved re-labeling all roughly 2,700 training examples and was performed by the two lead coders only.

Inter-annotator agreement was examined across 107 listings (a random stratified sample to achieve a sufficiently representative spatial distribution). Both lead coders separately annotated each document, and then each unique span tagged by either author was extracted (N=1,554). Treating one coder's tags as "true" and the other's as "predicted" yielded a weighted F1 score of 0.822 across all label categories. This measure of agreement is arguably conservative insofar as it requires exact span matches, e.g., "The MoMA" and "MoMA" would be treated as divergent predictions in the calculation of the F1 score.

## 4.2    Detecting spatial language

We evaluated three fine-tuned models on Distil-RoBERTa embeddings (Sanh et al., 2019; Liu et al., 2019) of our Airbnb text data: 1) linear classification, 2) a Conditional Random Fields (CRF) model (Lafferty et al., 2001) and 3) and a CRF-BiLSTM model (Huang et al., 2015). Additionally, we run a few-shot in-context learning experiment prompting ChatGPT to see how our custom models compare in performance to a larger LLM without fine-tuning. Models were given training data that had additionally been processed using IOB-chunking (Ramshaw and Marcus, 1999). With the DistilRoBERTa-CRF performing the best (with an F1-score of 0.814 on the validation set and 0.812 on the test set), we report all following downstream results with it (for a full comparison of models and details, see Appendix A).

## 4.3    Finding spatial dependencies

Listings frequently discuss distance and travel between places. These relations are essential to a full picture of toponymic reference. We call tokens

reflecting these relations Spatio-Temporal Entities (STEs). STEs such as "5 minutes from" or "walking distance" were tagged along with toponyms and other spatial entities. However, to move beyond a bag-of-words relationship between tags, we also parse dependencies between STEs and toponyms.

First, we label our corpus using the NER model described above. Next, we split each document into sentences and parse for dependencies between tokens using spaCy's transition-based dependency parser.[7] If a token also has a NER label, it is merged with any token belonging to the same IOB-chunk, inheriting all dependencies from its individual tokens. All tokens merged into entities and the remaining non-entity tokens effectively form a directed graph, which is filtered for any nodes that are labeled "STE"; all dependencies that point to these nodes are removed.

What remains is a set of weakly connected subgraphs that each have at most one STE node and $n$ nodes with other labels, including toponyms. If any of these nodes is a toponym, the STE must refer to it, yielding a final set of individual STE nodes and their dependent toponyms. This process is illustrated in Figure 2, where the sentence is initially one graph, that is then split into subgraphs at "Walking distance to" and "very close to", both with several dependent toponyms.

## 4.4    Resolving spatial language

To match spans that were tagged as neighborhoods outside of the canonical set to the selfsame, we develop a lightweight method for *toponymy resolution*. First, out of all neighborhood toponyms identified by our model but outside the canonical set, we keep those that are dependents of an STE (see subsection 4.3 and Figure 2), if and only if that STE is generally synonymous with "in," including expressions such as "in the heart of" (see

---

[7]https://spacy.io/api/dependencyparser

Appendix F for the full list). Second, the locations of the listings mentioning these toponyms are then used as input for a Kernel Density Estimation (KDE) model that filters out locations more than two standard deviations from the mean of the distribution. Thirdly, the remaining listings yield a convex hull for the span of each unique, "non-canonical" toponym. Finally, the $n$ nearest centroids of canonical neighborhood hulls are selected for closer analysis. Out of the $n$ nearest canonical toponyms, we next examine which are: (a) the $k$ nearest neighbors in terms of cosine similarity using both word2vec (Mikolov et al., 2013) and fastText (Bojanowski et al., 2017) models trained on the listing and review texts, and (b) the $m$ nearest neighbors in terms of Jaro-Winkler (Jaro, 1989; Winkler, 1990) distance, i.e. in terms of spelling similarity. Then, each neighborhood toponym outside the canonical set is assigned to the canonical toponym that scores best on these ensemble criteria. To validate the findings of this paper, we finally also went over this list and corrected it manually (for further details and F1-scores, see Appendix C)

## 5 Analysis

### 5.1 Toponymic self-reference

We begin with a simple demonstration that validates a relationship between neighborhood names and urban geospatial structure. To do so, we ask how frequently Airbnb listings in different neighborhoods mention their neighborhood by name, using toponymy resolution to capture misspellings and alternate usages. If such a relationship exists, we would expect listings in more central or otherwise desirable locations to invoke neighborhood names more frequently than those in less desirable areas. Figure 3 (a) plots this relationship, and indeed we see that this is generally the case: neighborhoods at the fringes of the city toponymically self-reference much less frequently than central neighborhoods. In other words, we begin to see that urban dynamics such as centrality and periphery are inscribed in "bottom-up" toponymic practices at scale. Furthermore, from Figure 1 we can see that listings also reference neighborhoods outside of their own location, generating a toponymic hierarchi of sorts.

These measures entail a major shortcoming, however: it assumes that the canonical neighborhoods we use align with the way Airbnb users imagine urban space. A host might in fact refer to their listing's location by neighborhood name, but they claim to be in a neighborhood other than what our geometries assume. This observation is suggestive of the idea of "vague cognitive regions" in geographical research: the variable ways that people categorize and break up geographic space, both cognitively and in discourse (Gao et al., 2017; Montello et al., 2014).

To investigate how cultural factors shape these cognitive regions, regardless of their relationship to canonical neighborhood boundaries, we proceed to a second analysis in which we use the spatio-temporal entity class of our NER model to induce *toponymic spans*.

### 5.2 Toponymic span

We analyze toponymic span by asking two questions: to what extent does the geospatial span of claims to neighborhood membership and proximity—e.g., *located in Midtown, close to Greenpoint*—differ from canonical neighborhood boundaries, and how does this vary as a function of neighborhood status?

This analysis makes use of two NER categories: spatio-temporal entities (STEs) and neighborhood toponyms. Using dependency parsing (Figure 2), we identify when a neighborhood toponym occurs as the syntactic child of any STE that indicate membership: *in the heart of*, *central to*, etc. (see Appendix F). This allows us to plot the geographic span of claims to neighborhood membership against the coordinates of corresponding Airbnb units.

To calculate a neighborhood's toponymic span, we take the convex hull of the points corresponding to membership claims, again employing KDE to remove any listings two standard deviations away from the distribution mean (for KDE details, see Appendix C). We take the area of the convex hull of this resulting set of coordinates, which can be compared to the area of the canonical neighborhood, as can be seen in Figure 3 (b).

Figure 4 visualizes this process in more detail for two neighborhoods: Williamsburg and Bedford-Stuyvesant ("Bed-Stuy"). The two neighborhoods are similar in many ways: both are quite large, contain high rates of Airbnbs, and are adjacent to one another. Both are centrally located to North Brooklyn and relatively well connected to the city by subway. While each has been the site of gentrification, attracting many young "transplants" moving to New York, Williamsburg is widely cited as one

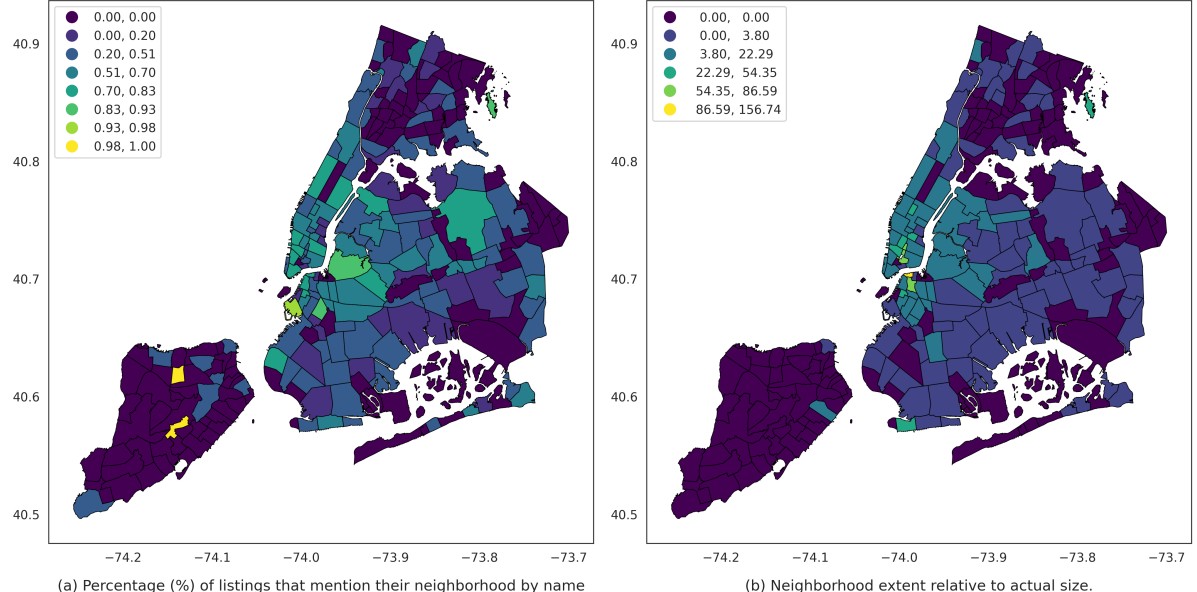

(a) Percentage (%) of listings that mention their neighborhood by name

(b) Neighborhood extent relative to actual size.

Figure 3: (a) Percentage of Airbnbs in each canonical neighborhood that refer to their neighborhood by name. The highest numbers are concentrated in particular parts of Brooklyn and Manhattan. The neighborhoods on Staten Island (in yellow, lower left corner) that have a very high ratio, also have only one or two listings. (b) The ratio between the extent of a neighborhood hull and its actual area.

of the most intense sites of gentrification in the city for two decades (see, e.g., Curran, 2007) and has been dubbed "the original hipster breeding ground" of 21st-century Brooklyn (Schiermer, 2014, 170). In this sense, Bed-Stuy lags behind, still attracting young transplants but with 25% lower median rents (Johnson et al., 2022, data appendix)[8] and generally lacking the desirability and fame (or notoriety) of Williamsburg.

These differences in prestige are revealed through practices of toponymic inscription at scale. The left column of Figure 4 displays *all* mentions of each neighborhood, without filtering for membership claims ("located in") or Kernel Density outliers. As we can see, Airbnb hosts across four city boroughs (excepting Staten Island) situate their locations with reference to Williamsburg, regardless of how they do so. By contrast, references to Bed-Stuy are much more locally concentrated, suggesting the lesser prestige attached to the neighborhood name. After filtering for usages that occur in the context of a membership claim and removing outliers with Mahalanobis filtering, the spans for both shrink considerably. Nevertheless, the ratio of the filtered span area to the underlying canonical

area is much higher for Williamsburg (6.929) than for Bed-Stuy (2.027), showing how local prestige can "stretch" an area's collective cognitive region.

While the comparison of these two neighborhoods serves as a useful demonstration, Figure 5 (panel 3) shows that the Pearson correlation between toponymic span and gentrification—operationalized with a state-of-the-art small-area index of gentrification (Johnson et al., 2022)—persists across the dataset ($r = .35$)[9]. The rightmost panel of Figure 5 displays the negative correlation between gentrification and toponymic reference *other than* neighborhood names. In other words, neighborhoods which have yet to gentrify invoke toponyms to situate their listings, but they are more likely to invoke other signposts—transit stations, businesses, or airports, for example—to communicate the location of their units.

## 6 Discussion

We suggest that critical toponymy is an underexplored theoretical framework with fruitful applications in applied natural language processing. Mining text data to discover place names is an old task in data science (e.g., Twaroch et al., 2008). Recent innovations in embedding-based models have greatly improved our ability to infer toponyms in

---

[8]Johnson et al. (2022) uses different canonical boundaries than we do. In their data, we define "Bed-Stuy" as all tracts in what they call Bedford and Stuyvesant Heights, and "Williamsburg" as all tracts in Williamsburg and North Side-South Side.

[9]We got similar outcomes using Local Outlier Factor, Mahalanobis distance, and Isolation Forest to filter span outliers.

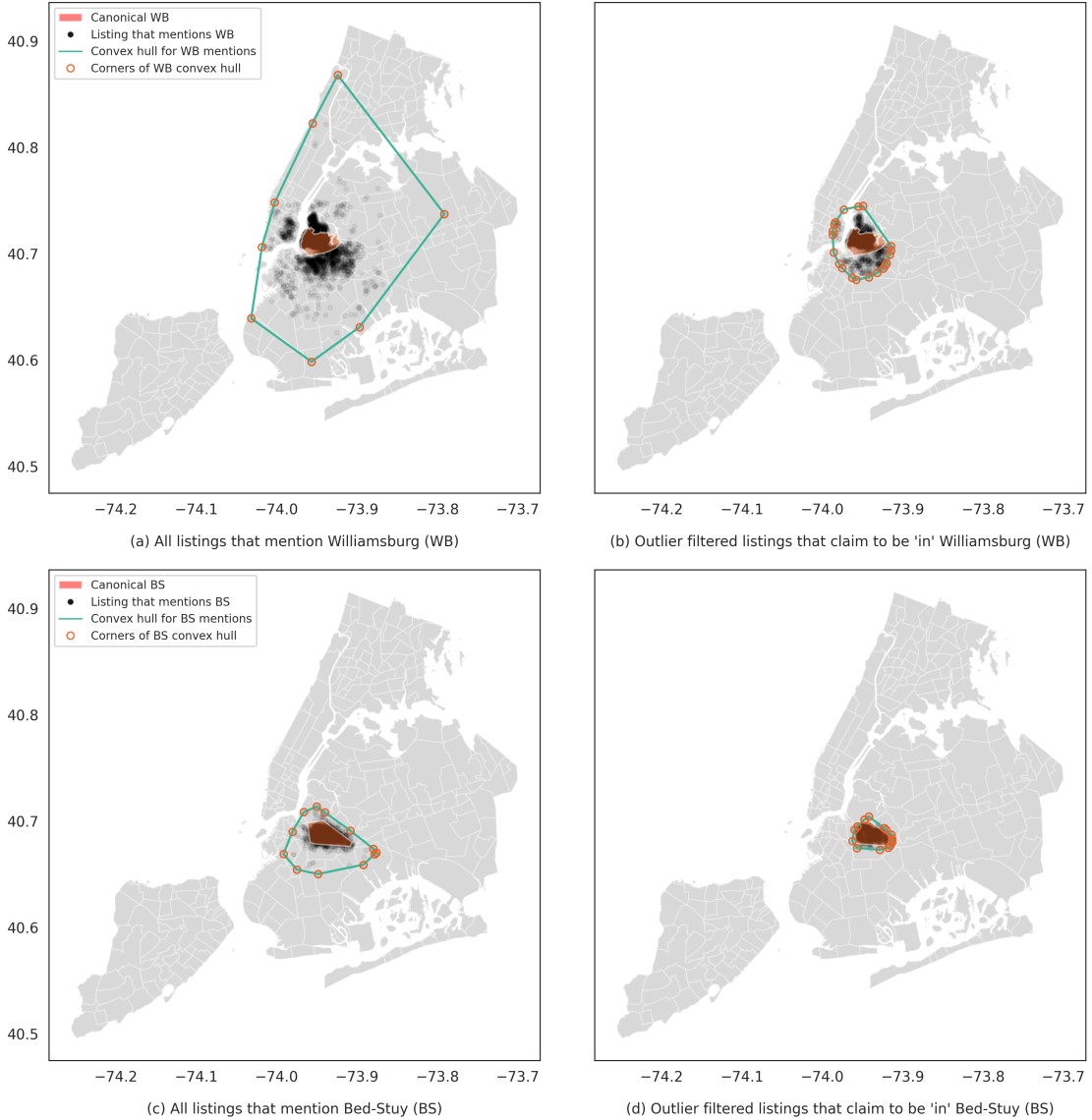

Figure 4: Visualizing toponymic span. The left column compares the unfiltered spans, defined as the complete complex hull area of all neighborhood mentions, of two adjacent neighborhoods: Williamsburg (a) and Bed-Stuy (c). Williamsburg's considerably larger total span suggests its perceived relevance to much of the city, whereas mentions of Bed-Stuy are more locally constrained. The right column depicts the convex hulls (Barber et al., 1996) after filtering mentions entailing claims of membership (e.g., *located in*) and after filtering with Mahalanobis distance to remove outliers. Again, Williamsburg's (b) greater ratio of its filtered span area to its canonical neighborhood area suggests its greater status as a neighborhood compared to Bed-Stuy (d).

unstructured text data. Thus far, such work in the context of toponymy has generally been explored as an engineering challenge to be improved upon with increasingly sophisticated methods (e.g., Cardoso et al., 2022; Davari et al., 2020; Fize et al., 2021; Tao et al., 2022). While such efforts are invaluable to engineers and social scientists alike, few have extended these novel approaches to concrete questions of social scientific intrigue.

Here we offer preliminary steps in this direction, introducing novel data and a bespoke NER model

to investigate the relationship between bottom-up toponymic practices and neighborhood status in the context of the Airbnb market. We demonstrate multiple ways in which the toponymic language reflects a variety of urban geospatial and sociocultural dynamics. Not all hosts locate their units in reference to their residential neighborhood. With some exceptions—an exploration of which would require a much more fine-grained discussion of the idiosyncrasies of New York City neighborhoods than is appropriate here—more peripheral neigh-

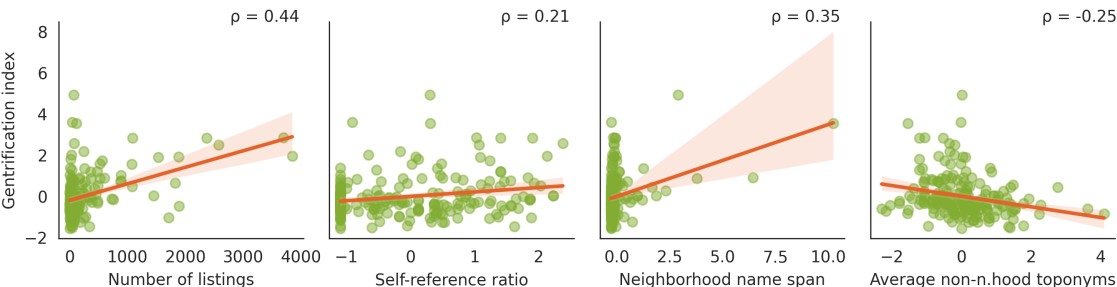

Figure 5: From left to right, using Pearson correlation: 1) The number of Airbnbs in a neighborhood positively correlates with the neighborhood's gentrification index, defined as in Johnson et al. (2022). 2) The proportion of Airbnbs that mention their neighborhood by name is also positively correlated with gentrification, as is the neighborhoods toponymic span (3). 4) On the other hand, toponymic references to categories other than neighborhood is negatively correlatted with gentification, suggesting that hosts turn to other toponymic resources when their neighborhoods lack cultural capital in the eyes of Airbnb consumers. Unlike in the natural sciences, Pearson correlation values in the ±0.2–0.5 range are regularly reported as substantively meaningful associations (e.g., Cohen, 1988).

borhoods are less likely to use neighborhood names to situate their locations, suggesting how urban dynamics of center and periphery come to be expressed in toponymic reference at scale.

Furthermore, we identify a relationship between gentrification dynamics and what we call *toponymy span*: the ratio between the area within which people claim membership in a neighborhood, and the area of its canonical boundaries. Given the demonstrated association between Airbnbs and gentrification (suggested by our data as well; see Figure 5, leftmost panel), this points to the possibility of a circular process. As a neighborhood gentrifies and acquires desirability among a class of largely white, middle- and upper-middle class, young professionals, Airbnb hosts and guests at the geographic fringes of those neighborhoods become more likely to locate themselves within it. Identifying themselves with new desirable toponyms might proceed to attract guests with greater efficacy, increasing the "rent gap" introduced by Airbnbs (Wachsmuth and Weisler, 2018) and perhaps accelerating the gentrification processes set in motion by STRs. While such causal dynamics would be difficult to properly model, and we certainly do not do so here, our findings in combination with prior research on the effect of Airbnbs suggest that toponymic practices might not merely reflect ongoing urban change, but play a more active role therein.

## 7 Future work

Due to space limitations, here we primarily focus on only two of the 21 categories our model

is trained to identify: neighborhood toponyms and spatio-temporal entities. Future work should investigate toponymic practices of different kinds. Larger text corpora, spanning longer periods of time, could reveal how these dynamics—for example, the relationship between toponym span and gentrification—play out diachronically. Many conventional indicators of gentrification process are measured with Census data that may lag behind the on-the-ground experience and economic effects of gentrification. If such linguistic signals could be shown to capture gentrification dynamics before they fully manifest in conventional sociodemographic data, the importance of toponymic analysis in urban contexts would become all the more apparent.

Finally, researchers in human-computer interaction could expand the scope of this research program. Qualitative analyses could add considerable depth to our understanding of how tourists, commuters, and prospective residents mobilize toponymic knowledge in the process of housing search practices, whether for short- or long-term rentals and home-buying. On the supply side, the same could be done for Airbnb hosts. Given the increasing professionalization of Airbnb hosting (Bosma, 2022; Dogru et al., 2020), the ways in which such practices vary across highly professionalized (and sometimes corporatized) Airbnb hosts compared with hosts who simply rent their spare rooms or apartments when they are away could merit attention as well.

## Ethical Considerations

**Data.** Our data comes from Inside Airbnb, which describes itself as *"a mission driven project that provides data and advocacy about Airbnb's impact on residential communities."*[10] Only public snapshots from Airbnb are collected and analyzed, and obfuscation is present, to a certain extent, for location information on listings. Despite the public nature of our data, it is unreasonable to assume that users explicitly consent for their data to be collected and analyzed in this way. As is with most computational social science research conducted at scale, it is infeasible to obtain explicit user consent for large-scale datasets such as ours (Buchanan, 2017). Here, we believe that the benefits of our work in illuminating the ways in which neighborhood dynamics are inscribed in toponymic practice outweigh its potential harms.

## Limitations

We interpret this paper as a proof-of-concept that relates theoretical perspectives from human geography and critical toponymy to NLP modeling, pointing towards numerous avenues for future research. In addition to the future work suggested above, our technical solutions could be improved in several ways. While our NER models achieved strong F1-scores, a larger annotated dataset would likely improve performance. Second, our results are specific to New York, and further work is required to determine to what extent they generalize to other cities. Third, we filtered out reviews and descriptions that were not estimated to be in English with a 0.95 or higher probability. Results might differ in sociologically substantial ways with a multilingual approach. Fourth, the data we use is unevenly distributed across space, an issue that future work could address by incorporating spatial smoothing techniques.

## Acknowledgements

Mikael Brunila has received funding from the Kone Foundation. Sky CH-Wang is supported by a National Science Foundation Graduate Research Fellowship under Grant No. DGE-2036197. We would like to thank the "NLP for Social Science" workshop participants from McGill and Columbia for their thoughtful feedback on preliminary results.

---

[10] http://insideairbnb.com/about/

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

| Model | F1 | Precision | Recall |
|---|---|---|---|
| DistilRoBERTa-LC | 0.807 | 0.784 | 0.832 |
| DistilRoBERTa-CRF | **0.814** | **0.803** | 0.826 |
| DistilRoBERTa-CRF-BiLSTM | 0.812 | 0.789 | **0.836** |
| ChatGPT w. In-Context Learning | 0.398 | 0.506 | 0.328 |

Table 2: Among the four NER architectures implemented, DistilRoBERTa with CRF layer performed the best on the validation set.

## A  NER Models

Table 2 shows the performance of the top model for each implemented architecture: (1) DistilRoBERTa with a linear classifier, (2) with a CRF classifier, (3) with a CRF-BiLSTM classifier, and (4) ChatGPT with In-Context Learning. We ran all experiments using both the initial set of annotations as well as the second, corrected set of annotations. All custom models were trained over five epochs with 2718 total examples, a 80/10/10 train-test-validation split, a $1 \times 10^{-4}$ learning rate, $1 \times 10^{-5}$ weight decay, gradient clipping, and early stopping. A grid search was done over dropout values from 0 to 0.3, batch sizes from 4 to 32, and a hidden layer size from 100 to 400. Overall there was not a great difference in model performance within each architecture.

Among the models, DistilRoBERTa with a CRF layer performed best, while ChatGPT lagged behind even with in-context examples. For the former architecture, the best performing model ran for five epochs, with a batch size of four, dropout of 0.3, and 300 hidden layers. The performance of this model on the test set and individual NER tags can be seen in Table 3. In addition to the tags in the table, the dataset contains the tag "TN-OTHER", which was ultimately excluded from the training process due to its infrequency and irrelevance.

| | Label | N Annotated (all) | N Predicted (listings) | N Predicted (reviews) | F1 | Recall | Precision |
|---|---|---|---|---|---|---|---|
| 1 | TN:AIRPORT | 441 | 5839 | 4637 | 0.877 | 0.909 | 0.847 |
| 2 | TN:BEACH* | 46 | 361 | 147 | 0.609 | 0.538 | 0.7 |
| 3 | TN:BOROUGH | 2005 | 30599 | 45236 | 0.974 | 0.970 | 0.978 |
| 4 | TN:BRIDGE_TUNNEL | 81 | 1688 | 1091 | 0.375 | 0.5 | 0.3 |
| 5 | TN:BUSINESS | 948 | 22720 | 9760 | 0.832 | 0.832 | 0.832 |
| 6 | TN:CITY | 1223 | 20601 | 53001 | 0.890 | 0.906 | 0.875 |
| 7 | TN:HOSPITAL | 85 | 1021 | 219 | 0.769 | 0.833 | 0.7142 |
| 8 | TN:NAT_FEAT | 64 | 2239 | 543 | 0.667 | 0.714 | 0.625 |
| 9 | TN:NEIGHBORHOOD | 2792 | 67430 | 45131 | 0.937 | 0.955 | 0.919 |
| 10 | TN:PARK | 578 | 17224 | 9454 | 0.946 | 0.984 | 0.910 |
| 11 | TN:REGION | 49 | 332 | 655 | 0.462 | 0.375 | 0.6 |
| 12 | TN:SCHOOL | 158 | 3491 | 1128 | 0.837 | 0.857 | 0.818 |
| 13 | TN:STATION | 464 | 17124 | 6624 | 0.696 | 0.722 | 0.672 |
| 14 | TN:STREET | 672 | 22623 | 8795 | 0.706 | 0.712 | 0.700 |
| 15 | TN:TOURIST_ATTR | 812 | 19151 | 6228 | 0.713 | 0.662 | 0.773 |
| 16 | GEOG_ENTITY | 9024 | 190203 | 289271 | 0.827 | 0.833 | 0.821 |
| 17 | HOST_BUILDING | 8122 | 277925 | 12391 | 0.733 | 0.762 | 0.705 |
| 18 | TRANSIT | 4998 | 97983 | 108324 | 0.840 | 0.843 | 0.837 |
| 19 | SPAT_TEMP_ENT | 11485 | 358506 | 203545 | 0.789 | 0.822 | 0.758 |
| 20 | WALK_RUN_BIKE | 128 | 2915 | 7368 | 0.552 | 0.571 | 0.533 |
| | **Overall** | **26777** | **769150** | **756132** | **0.812** | **0.830** | **0.795** |

*No tags in test set, F1, recall, precision results from validation set.

Table 3: Summary of NER label frequencies in the training data and the overall data, as well as performance metrics (F1, recall, and precision) for the DistilRoBERTa-CRF model on the test set.

| Index | Toponym | Frequency |
|-------|---------|-----------|
| 1 | spanish_harlem | 164 |
| 2 | metropolitan_museum_of_art | 162 |
| 3 | domino_park | 152 |
| 4 | madison_avenue | 150 |
| 5 | wholefoods | 129 |
| 6 | robertas | 120 |
| 7 | cloisters | 114 |
| 8 | brooklyn_public_library | 105 |
| 9 | north_williamsburg | 104 |
| 10 | bedford_l | 104 |

Table 4: The sample from the top toponyms that were predicted by the DistilRoBERTa-CRF NER model but that were not present in the training data.

## B  Out Of Data Performance

To demonstrate the capacity of our NER model to generalize beyond the training set, we look at some of the toponyms the model predicted from the rest of the Airbnb data. In total, the model finds $97,908$ toponyms that were not in the train set but were among the other listings and reviews. These include well-known museums, alternative neighborhood names ("Spanish Harlem" for East Harlem), partial areas of neighborhoods ("North Williamsburg"), and popular parks and restaurants, all with a fairly high frequency in the data. While we did not perform extensive experiments on this task, Table 4 shows some examples from the most common toponyms predicted by the model, providing some preliminary evidence that the model is indeed able to generalize broadly.

## C  KDE & Toponymy Resolution

All Kernel Density Estimation (KDE) models in this paper are fitted using the default models in scikit-learn.[11]. Specifically, we use a Gaussian kernel with Euclidean distance and a bandwidth of $1.0$.

For the toponymy resolution pipeline, we consider the 20 nearest centroids, 100 nearest neighbors for both word2vec and fastText as well as the 100 nearest Jaro-Winkler neighbors. Some spans tagged as neighborhoods fall outside of the pipeline, because they are not used sufficient times to form a convex hull. All in all, this pipeline achieves a precision of $0.745$, a recall of $0.984$, and F1-score of $0.848$. These results were calculated after first running the model, then correcting the pairing of canonical and non-canonical toponyms manually, and finally, comparing the original model output with the manually corrected results.

## D  Expanded Dataset Details

The following table describes each category of entity in our NER model. The prefix TN indicates that a category is toponymic rather than generic.

A challenge to labeling training data is that many tokens could fall into multiple categories. There are at least two common reasons for this. One is that the word or phrase refers to different things in different contexts. For example, many subway stations are named for their street or neighborhood (e.g., *Forest Avenue M station*). In situations such as these, coders whole label the whole phrase as TN:STATION rather than labeling *Forest Ave* as TN:STREET, since the author was using it in the context of a named subway station. A second source of ambiguity is that a word or phrase could reasonably be placed into multiple categories. For example, Central Park is both a park and a tourist attraction. In situations such as these, coders opted for the most specific category (TN:PARK in this example).

---

[11]https://scikit-learn.org/stable/modules/generated/sklearn.neighbors.KernelDensity.html

| Category | Definition | Examples |
|---|---|---|
| TN:NEIGHBORHOOD | Any named neighborhood | Soho, UWS |
| TN-BOROUGH | Any named borough | New York City, Jersey City |
| TN:CITY | Any named city | Manhattan, Bk |
| TN-REGION | Any named area larger than a city | the Hamptons, Upstate |
| TN:STREET | Any named street | Broadway, Fifth Ave |
| TN:BRIDGE_TUNNEL | Any named bridge or tunnel | Brooklyn Bridge, Verrazano |
| TN:STATION | Any named transit center | Grand Central, Penn Station |
| TN:AIRPORT | Any named airport | JFK, LaGuardia |
| TN:PARK | Any named park | Central Park, Riverside |
| TN:SCHOOL | Any named educational institute | Columbia, NYU |
| TN:BUSINESS | Any named business | Dunkin, CVS |
| TN:HOSPITAL | Any named business | Dunkin, CVS |
| TN:TOURIST_ATTR | Any named tourist museum, landmark, etc. | the Met, Statue of Liberty |
| TN:BEACH | Any named beach | Jacob Riis Park, Rockaways |
| TN:NAT_FEAT | Any named natural feature | Atlantic, East River |
| TN:OTHER | Toponyms poorly captured in other categories | zip codes |
| GEOG_ENTITY | Any generic institution or other entity in the environment | grocery stores, beaches |
| TRANSIT | Any generic reference to transit options | F train, the bus |
| WALK_RUN_BIKE | Any generic reference to walking, running, biking as leisure activities | walk, bike |
| SPAT_TEMP_ENT | Any expression of spatiotemporal relation | 25 minutes from, 3 blocks away |
| HOST_BUILDING | Any reference to the host's unit as a whole, or to amenities associated with the property but outside of the unit | 25 minutes from, 3 blocks away |

## E   Complete Annotation Guidelines

This document describes the process for annotating Airbnb listings and comments in order to create training data for a named entity recognition (NER) model. The goal is to use this model to automatically extract many sorts of references to place in these listings. There are 15 categories of spatial reference, and the rules for identifying which is appropriate are listed below. Before detailing the categories, there are three universal ground rules for determining annotations:

- **Always omit determiners from noun phrases.**

    - i.e., tag Metropolitan Museum of Art, not The Metropolitan Museum of Art; grocery stores instead of a few grocery stores; corner of 29th and 6th instead of the corner of 29th and 6th. Inconsistency with this will harm model performance.

- **If a noun phrase contains an adjective, omit the adjective unless it is an essential part of the noun phrase.**

    - "Essential" introduces a bit of interpretation, but here are some guidelines: tag "Mexican restaurant," "grocery store", or "cocktail bar", but omit the adjectives in phrases like "nice restaurants," "24/7 stores", or "fun bar".

- **Do not tag any references to the interior of the Airbnb unit itself.**

    - We have a category, HOST_BUILDING, which is for references to certain building features, but we don't want to tag things like bathroom attached to the bedroom, or anything else *inside* of the apartment.

**Annotation Categories.**    There are 11 categories that start with the prefix **TN**. This stands for toponym, and it means that the reference to a location or spatial relationship involves directly referencing a specific identity by name. *Walgreens*, *the Hudson River*, *The Whitney Museum*, *Central Park*, *5th Avenue*, *NYU*, and *Brooklyn* are all toponyms. TN categories are listed below, in no particular order:

- TN:NEIGHBORHOOD

– These should be mostly captured automatically by our list of neighborhood names, but things like misspellings might require human annotation.
– If the text modifies the neighborhood with words like *downtown/uptown* or *upper/lower*, include those words in the description (i.e., *Downtown Flushing*, *Lower Manhattan* should be tagged as TN:NEIGHBORHOOD)

- TN:STATION

  – TN:STATION refers specifically to specific transit stations (e.g., *the Bergen 2/3 stop*, *Myrtle-Willoughby Station*, *Grand Central Station*)
  – References to subway, bus, ferry lines *without mentioning a specific stop* (e.g., *the 6 train*) should be tagged as TRANSIT, **not** TN:STATION
  – Should include well known stations that could potentially be TN:TOURIST_ATTR (e.g. Grand Central) and train stations (e.g. Penn Station)

- TN:CITY

  – This will almost always be New York City, NYC, New York/NY (when clearly referencing the city and not the state)

- TN:BOROUGH

  – Any reference to a borough. These should mostly be automated, but some misspellings or shorthands (e.g., BK for Brooklyn) may need to be done by hand

- TN:PARK

  – Any named park: Central Park, Prospect Park, Greenwood, etc. Some neighborhoods in NYC have the word Park in them (e.g., Ozone Park), so make sure to double check if you are unsure.

- TN:SCHOOL

  – These will mostly be colleges and universities—Columbia, NYU, Fordham, CUNY, etc.—though references to other schools should also be marked with this.

- TN:TOURIST_ATTR

  – Primarily museums, performance venues, and landmarks (Statue of Liberty). Other things that could be considered tourist attractions, such as universities or famous stores, should be marked in those respective categories rather than the more general TOURIST_ATTR.
  – Ambigious cases that should be tagged here:
    * The Highline
    * Apollo Theater, etc.
    * All botanical gardens and zoos

- TN:STREET

  – Street names; street corners (e.g., "29th and 5th Ave" should be split up ("29th" and "5th Ave" as separate street names)
  – Include "square" and "Sq."
  – Other ambiguous cases that should be tagged here include Columbus Circle.

- TN:BUSINESS

  – Any named business: CVS, Walgreens, Domino's, Macy's etc.

- TN:OTHER

  – This is a grab-bag category.

- Zip codes

- **TN:AIRPORT**

  - Any named reference to JFK, LGA, EWR, or other airports

- **TN:REGION**

  - Any toponymic area larger than a city: counties, subregions ("The Hamptons"), states, countries
  - "NY" can be ambiguous as to the city or the state; use judgment according to sentence of context

- **TN:BEACH**

  - Any toponymic beach. The difficulty here is that many beaches also share a name with the neighborhood. The coder will have to use judgment to determine whether it is the beach itself or the neighborhood being referenced. We expect the majority of usages to refer to the beach

- **TN:BRIDGE_TUNNEL**

  - Any toponymic reference to bridges and tunnels (e.g., Holland Tunnel, Williamsburg Bridge). Brooklyn Bridge, although a tourist attraction, should be labeled as TN:BRIDGE_TUNNEL, not TOURIST_ATTR

- **TN:HOSPITAL**

  - Any toponymic reference to hospitals. Include university hospitals (e.g., Columbia University Medical Center/CUMC)

- **TN:NAT_FEATS**

  - Forests, named places for hiking, mountains, etc.
  - Bodies of water (e.g. Hudson River, the East River)
  - NOT man-made parks inside the city; that would be TN:PARK

The remaining categories do not refer to specific, unique, named entities, but generally to categories of things.

- **SPAT_TEMP_ENT**

  - Short for spatio-temporal entity. This is probably the most frequent category, and it is used to denote words and phrases describing proximity, distance, adjacency, location, and so on.
  - In general, these will be variations on prepositional phrases: *located on the corner of, situated at the intersection of, just a ten minute walk from, five minutes by bike to, less than 20 minutes from, two subway stops away, within walking distance, only minutes away, between 9th and 10th ave, easy access to*
    * When distances and times are modified (e.g., *roughly* 20 minutes away, *just* around the corner), include these modifying words
  - Occasionally could be adjectives: in a *central* location
  - Include "view of", "see all of" (e.g. *see all* of Manhattan from my window)
  - If a mode of transit is invoked

- **TRANSIT**

  - Non-toponymic references to transit. This will often be names of subway and bus lines without the stop (*the 1 train; nearby the F / G lines*) or to the mode of transit in general: *subway, subway stop, bus, bus stop, PATH*, etc.

- **WALK_RUN_BIKE**

- Walking, running and biking as an activity that's not covered by STE

• GEOG_ENTITY

- Also one of the most common categories. Geographic entities comprise most other generic references to certain places/establishments. Often these will be "things to do": *parks, restaurants, bars, shopping malls,* etc. Other times it will be more abstract: *in a quiet neighborhood, on a bustling street, in a residential area*
- Trees, street parking
- Tag "community" here if it's synonymous to neighborhood. (e.g. *East Harlem is still quite a poor community*.)

• HOST_BUILDING

- References to the building in which the unit is located. This can be a reference to the building itself (l*ocated in a historic brownstone*), or to spatialized amenities of the building but outside the unit itself: *guest access to the rooftop terrace,*
- OR anything that implicitly refers to the building in its entirety (*the room is close to the subway*)
- OR, references to amenities associated with the building that are spaces/that one can spend time in: backyards, roofs, patios, gyms (but not washing machines, kitchens, or anything else INSIDE the apartment itself)
- OR interfaces between world and building (e.g. *there is a secure door with a code*)
- Includes features such as "stoop"

## F  Inductively generated STE set indicating membership

- in
- in the heart of
- located in
- located in the heart of
- right in the heart of
- in the middle of
- in the center of
- centrally located in
- located in the center of
- conveniently located in
- nestled in the heart of
- right in the middle of
- situated in the heart of
- at the heart of
- conveniently located in the heart of
- located right in the heart of
- nestled in
- ideally located in
- centrally located in the heart of
- located in heart of
- located in the middle of
- tucked away in
- perfectly located in
- located in the epicenter of
- in the very heart of
- located in the best part of
- perfectly located in the heart of
- located right in the middle of
- in the epicenter of
- in the
- located within
- remarkably located in the heart of
- absolute heart of
- close to the heart of
- located in center of
- at the center of
- in center of
- in this part of
- centrally located on
- ideally located in the heart of
- in the hub of
- right at the heart of
- very right in the middle of
- located at the center of
- located on the heart of
- truly in the heart of
- located in the western section of
- in a great part of
- at the nexus of
- conveniently located in the middle of
- within the heart of
- very centrally located in
- conveniently situated in
- within walking distance from the heart of
- located in the heart
- steps away
- in the south side of
- perfectly situated in the heart of
- set in the heart of
- located in a very convenient area of
- very conveniently located in
- in the very center of
- locate in
- perfectly situated on
- best part of
- in the midst of
- steps away to
- short walk away from the heart of
- in a great location in
- primely located in
- inside
- convenient location in central area of

- in a prime position in
- just a few steps away from
- in the most desirable part of
- in the lower part of
- within steps of
- footsteps away from
- located in the south of
- in the best part
- located in the midst of
- in the heat of
- located in a part of
- conveniently in
- by the heart of
- in middle of
- located at the nexus of
- in the coolest part of
- in the northernmost portion of
- only steps away from
- located conveniently in
- literally steps away from
- strategically situated at the center of
- located right in
- nestled in the best part of
- in the prime location of
- convenient in
- perfectly placed in the middle of