# OpenReview forum: "Toward a Critical Toponymy Framework for Named Entity Recognition: A Case Study of Airbnb in New York City"
_EMNLP/2023/Conference — EMNLP 2023 Main_

### Official Review · Reviewer_B1cM · 2023-08-01

**Typos Grammar Style And Presentation Improvements:** F1-score and Appendix number are miss…
**Soundness:** 4

**Excitement:**

4: Strong: This paper deepens the understanding of some phenomenon or lowers the barriers to an existing research direction.

**Paper Topic And Main Contributions:**

In this work, a novel NER model is proposed to identify both formal and informal toponyms as well as spatial relationships in unstructured text from Airbnb listings. For this purpose, the authors develop a taxonomy of 21 toponymic categories, including spatio-temporal relations ("5 min away from...") and manually annotate 2600 listings. Three fine-tuned transformer models are compared, with the CRF performing the best and presenting a good F1 score.

This model is then used to analyze the NYC Airbnb dataset in relation to a gentrification index. It is found that the proportion of Airbnbs listings that mention its neighborhood my name is positively correlated with gentrification, as is the neighborhood toponomyc span, suggesting that the perceived "prestige" of a neighborhood creates a tendency for its name to be referenced in a larger surrounding area. Inversely, toponomyc references to entities other than neighborhood are inversely correlated with gentrification.

**Reasons To Accept:**

This work focuses on a sociological topic that is both highly relevant and of general interest: the phenomenon of gentrification in big cities such as NY. The work makes use of modern NLP techniques (transformers, embeddings, fine-tunning) to improve the state of the art in toponymic NER, showing convincingly that it can identify informal toponyms. This seems particularly important in a context where the perceived boundaries and names of neighborhoods are an emergent collective phenomenon, nevertheless of strong sociological interest, impacting the perception of prestige of neighborhoods and affecting / being affected by the dynamics of gentrification.

I really appreciate the successful inter-disciplinary nature of this work, which is a great example of high quality computational social science. Beyond the soundness of the NLP methods proposed and the improvement in this specific domain, interesting sociological observations are actually extracted, in the correlations found between gentrification and informal toponomycal practices.

**Reasons To Reject:**

This method was tested on a single city (NY) of which the authors appear to have good knowledge. It remains to be seen how well it generalizes to other cities.

**Reproducibility:**

5: Could easily reproduce the results.

**Reviewer Confidence:**

3: Pretty sure, but there's a chance I missed something. Although I have a good feel for this area in general, I did not carefully check the paper's details, e.g., the math, experimental design, or novelty.

---

> ### Author Rebuttal · Authors · 2023-08-28
>
> The authors would like to thank all three reviewers for their thoughtful, constructive, and knowledgeable feedback. We appreciate the opportunity to offer our responses here. Below, we respond to each reviewer’s “Reasons to reject” and “Questions for the authors” sections. For the sake of brevity we do not respond to “Reasons to accept,” “Missing references,” and “Typos” feedback; reviewers can assume that we will incorporate all suggested references and correct all typos in the final draft. We reproduce each reviewer comment in quotation marks below.
>
> Reasons to reject
>
> “This method was tested on a single city (NY) of which the authors appear to have good knowledge. It remains to be seen how well it generalizes to other cities.”
>
> We agree that, in an ideal world, we would be able to annotate data from multiple cities and test performance on data from unseen cities. Due primarily to practical limitations, data from other cities were not included in this paper. We argue that this paper is proof-of-concept for a general way of using a class of NLP methods along with social science theory to think about a certain kind of linguistic practice and its relationship to urban social life, in line with the purview of the “Computational social science and cultural analytics” track. As such, we feel that a single city is sufficient for the substantive argument we make, if not ideal.

---

### Official Review · Reviewer_AMJs · 2023-08-04

**Soundness:** 4

**Excitement:**

4: Strong: This paper deepens the understanding of some phenomenon or lowers the barriers to an existing research direction.

**Missing References:**

Section 2.2 could be improved by surveying more relevant works in toponym resolution from the NLP literature. The following references can be a starting point:

DeLozier, G., Baldridge, J. and London, L., 2015, February. Gazetteer-independent toponym resolution using geographic word profiles. In _Proceedings of the AAAI Conference on Artificial Intelligence_ (Vol. 29, No. 1).

Gritta, M., Pilehvar, M.T., Limsopatham, N. and Collier, N., 2018. What’s missing in geographical parsing?. _Language Resources and Evaluation_, 52, pp.603-623.

Kamalloo, E. and Rafiei, D., 2018, April. A coherent unsupervised model for toponym resolution. In _Proceedings of the 2018 World Wide Web Conference_ (pp. 1287-1296).

Schneider, N.R. and Samet, H., 2021, November. Which Portland is it? a machine learning approach. In _Proceedings of the 5th ACM SIGSPATIAL International Workshop on Location-based Recommendations, Geosocial Networks and Geoadvertising_ (pp. 1-10).

**Paper Topic And Main Contributions:**

The paper presents a framework for critical toponymy that aims to study place names (toponyms) within a broader social, cultural, historical, and political context. To this end, the authors curated a dataset comprising nearly 47K New York Airbnb listings and reviews. They have then undertaken the manual annotation of toponyms within nearly 2.6K of these listings and reviews based on a taxonomy they devised. Subsequently, an NER model was trained on the annotated data and was used to identify toponyms in the rest of the dataset. A statistical toponym resolution model is introduced to ground toponyms using syntactic features (dependency parse tree connections) and semantic similarity in the embedding space (word2vec and fastText).

Given the toponymy information, the paper provides empirical support to corroborate a number of sociodemographic phenomena:  (1) users exhibit a tendency to use neighborhood names more frequently for neighborhoods that are central and deemed desirable (Figure 2); (2) A disparity between the perceived boundaries of neighborhoods and their canonical boundaries is underscored in two contrasting neighborhoods in terms of prestige: Williamsburg and Bedford-Stuyvesant (Figure 3); (3) The correlation of several factors with gentrification index is visualized (Figure 4). The number of listings, the ratio of self-referential mentions to neighborhoods, and the span of neighborhood names exhibit positive correlations with gentrification.


**Questions For The Authors:**

A. At the final step of the lightweight toponym resolution algorithm $\S$4.4, how is the number of points determined to rank the candidates?

B. It is not clear precision/recall/F$_1$ corresponds to what data in Table 2.


**Reasons To Accept:**

1. The paper explores an interesting and under-studied topic in the NLP community.
2. The insights in the paper could spark future work to bridge the gap between NLP and other disciplines such as social studies.
3. The manual data collection process is thorough and well-executed, especially the curated taxonomy of toponyms.
4. The paper is well-written and I really enjoyed reading it.


**Reasons To Reject:**

I do not find any major reasons to reject this paper. Nevertheless, I do have a few minor concerns:

1. The case study of toponym spans for two neighborhoods is interesting, yet, we cannot draw concrete conclusions based solely on one instance. A broader approach would be to conduct a similar analysis on all gentrified and non-gentrified neighborhoods, which provides a more robust base for hypothesis testing. This does not mean visualizing all cases, but the area of the estimated convex hulls can be used for comparison.

2. The correlations depicted in Figure 4, while present, appear to be moderate. It is not clear what correlation metric is used. My guess is the Pearson coefficient. In that case, 0.37 and 0.44 may not attain a level of correlation deemed substantial.

3. The performance of the lightweight toponym resolution method is not evaluated. An analysis is required to show how accurately this algorithm performs.

4. The performance of the NER model on TN:OTHER and TN:REGION in Table 2 stands out as notably low. For TN:OTHER, it might be attributed to the small number of annotations, but the reasons behind the low performance on TN:REGION is not clear.


**Reproducibility:**

4: Could mostly reproduce the results, but there may be some variation because of sample variance or minor variations in their interpretation of the protocol or method.

**Reviewer Confidence:**

3: Pretty sure, but there's a chance I missed something. Although I have a good feel for this area in general, I did not carefully check the paper's details, e.g., the math, experimental design, or novelty.

**Typos Grammar Style And Presentation Improvements:**

- The paper is verbose at times. For instance, the algorithm described in $\S$4.4, could be illustrated as a pseudo-code. Similarly, Section 6 mostly reiterates the findings presented in earlier sections. By condensing some sections, it would be possible to accommodate important details like Table 1 in the main body.
- L320 and L322: XX
- L364: Appendix C $\rightarrow$ Appendix E
- Some citations include an unrelated number, e.g. L88, and L135.

---

> ### Author Rebuttal · Authors · 2023-08-28
>
> The authors would like to thank all three reviewers for their thoughtful, constructive, and knowledgeable feedback. We appreciate the opportunity to offer our responses here. Below, we respond to each reviewer’s “Reasons to reject” and “Questions for the authors” sections. For the sake of brevity we do not respond to “Reasons to accept,” “Missing references,” and “Typos” feedback; reviewers can assume that we will incorporate all suggested references and correct all typos in the final draft. We reproduce each reviewer comment in quotation marks below.
>
> Reasons to reject
>
> “The case study of toponym spans for two neighborhoods is interesting, yet, we cannot draw concrete conclusions based solely on one instance. A broader approach would be to conduct a similar analysis on all gentrified and non-gentrified neighborhoods, which provides a more robust base for hypothesis testing. This does not mean visualizing all cases, but the area of the estimated convex hulls can be used for comparison.”
>
> We agree that a two-neighborhood comparison is insufficient for more general inference. In a sense, this is the goal of Figure 4, panel 3, in which we show a correlation between the gentrification index and neighborhood span. If accepted, however, we will consider supplementary ways to demonstrate and visualize this relationship across all neighborhoods.
>
> “The correlations depicted in Figure 4, while present, appear to be moderate. It is not clear what correlation metric is used. My guess is the Pearson coefficient. In that case, 0.37 and 0.44 may not attain a level of correlation deemed substantial.”
>
> We will clarify that we are using Pearson’s correlation. We would argue that in the social science literature, correlations around 0.4 are commonly reported and generally considered meaningful.
>
> “The performance of the lightweight toponym resolution method is not evaluated. An analysis is required to show how accurately this algorithm performs.”
>
> The first reviewer said similarly and we will provide a comparison in an appendix (see our response above for further details).
>
> “The performance of the NER model on TN:OTHER and TN:REGION in Table 2 stands out as notably low. For TN:OTHER, it might be attributed to the small number of annotations, but the reasons behind the low performance on TN:REGION is not clear.”
>
> Both categories are quite low-N and we anticipate that this is why they perform poorly. We will clarify this.
>
> Questions for the authors
>
> “A. At the final step of the lightweight toponym resolution algorithm § 4.4, how is the number of points determined to rank the candidates?”
>
> We are not completely sure we understand the question correctly, but a candidate gets one point if it’s within the n nearest neighbors for fastText and word2vec and two points if it’s among the top n Jaro-Winkler neighbors. We arrived at this ranking system through empirical trial-and-error. Hopefully this addresses the question? To clarify how performance depends on the ranking system, we will compare different point ranking systems (only one point for Jaro-Winkler, different sizes of n etc) in the appendix, alongside performance comparisons with Mordecai recommended by reviewer QGQw.
>
> “B. It is not clear precision/recall/F corresponds to what data in Table 2.”
>
> We will clarify this in a final draft.

---

### Official Review · Reviewer_QGQw · 2023-08-05

**Soundness:** 4

**Excitement:**

4: Strong: This paper deepens the understanding of some phenomenon or lowers the barriers to an existing research direction.

**Missing References:**

There's a recent paper that discusses current challenges associated to text geoparsing, which should be analyzed in the context of the parts that concern "detecting spatial language" and "resolving spatial language":

Geoparsing: Solved or Biased? An Evaluation of Geographic Biases in Geoparsing
https://agile-giss.copernicus.org/articles/3/9/2022/

There's also paper describing a benchmark platform named EUPEG for evaluating text geoparsing over multiple datasets, which can perhaps be referenced in the analysis of related work:

Enhancing spatial and textual analysis with EUPEG: an extensible and unified platform for evaluating geoparsers https://geoai.geog.buffalo.edu/EUPEG/
https://github.com/geoai-lab/EUPEG

Another recent system for toponym recognition and disambiguation is Mordecai, which is conveniently available as open-source software. Passing texts through Mordecai and comparing results against the proposed approaches for "detecting spatial language" and "resolving spatial language" should be relatively simple (although this comparison would not use the detailed categories used in the proposed NER system):

Mordecai 3: A Neural Geoparser and Event Geocoder
https://arxiv.org/pdf/2303.13675.pdf
https://github.com/ahalterman/mordecai3

**Paper Topic And Main Contributions:**

The paper combines critical toponymy with automated text analysis. Using a dataset of 47,440 New York City Airbnb listings from the 2010s, the authors report on an study that focuses on how property owners refer to places within the city and how this relates to cultural and economic capital. The authors introduce a new Named Entity Recognition (NER) model tailored to identify places and spans of text that are important to the characterization of place, and also new lightweight and accurate methods for toponymy resolution and geospatial dependency parsing. Results show that the residents of different neighborhoods indeed use different strategies to describe their property locations, and that strategies such as mentioning the neighborhood by name positively correlate with gentrification. The authors also highlight the fact that while some previous work within the NLP community has focused on toponym recognition and resolution, few studies have actually been reported using these methods to address concrete questions related to other fields (e.g., human geography and/or the computational social sciences).

**Questions For The Authors:**

A - Section 4.1 should ideally report on the agreement between the annotators of the NER data, even if only leveraging on a small sub-set of the annotated data.

B - Section 4.2 mentions that the DistilBERT-CRF model performed best in the NER experiments, but the section does not present the actual F1 metric (i.e., in states "an F1-score of XX"). The authors should also explain why they haven't considered using a larger BERT model for producing the representations, or why they haven't considered fine-tuning the DistilBERT model directly for the NER task (e.g., the "linear classifier" approach seems to correspond to the training of a linear layer on top of the BERT representations for each token, and this setting could be easily extended to consider also the training of the DistilBERT model itself). Ideally, additional technical details about the training and inner-workings of the model(s) should also be given. For instance, what were the hyper-parameters considered for model training (this is only shown in Appendix A, but it is somewhat strange that different values for parameters such as the batch size were used for the different models)? How was span recognition modeled as a token-classification problem in terms of tags associated to the tokens (e.g., did the authors use a B-I-O tagging scheme, or some other approach)? How was the BERT tokenization scheme, based on sub-words, combined with the tokenization scheme used for creating the annotated data?

C - On what regards Section 4.4, the authors mention the use of a Kernel Density Estimation procedure to filter out locations more than three standard deviations from the mean of their coordinates. This procedure should be presented in more detail (e.g., what KDE parameters, such as the kernel function or the kernel bandwidth, were considered? What do the authors mean with the expression "mean of their coordinates"?), and the authors should also justify this choice against some other procedure to remove outliers. In connection to this, the authors also mention the use of a convex hull to delineate the region formed by the coordinates, but a more accurate procedure could have been used instead (e.g., threshold the results of the KDE function, to keep the high density region, or using a different computational geometry approach such as alpha-shapes, which is not restricted to the formation of convex polygons).

D - Also on what regards Section 4.4, which discusses methods for resolving spatial language, the authors mention the use of a "canonical list of toponyms." This should be further detailed in the paper. Does the canonical list correspond to a sub-set of geonames entries? The entire contents of Section 4.4 are actually somewhat confusing (e.g., it is also not clear why and how different string similarity metrics are used), and perhaps this section should be re-written.

E - The models used for detecting spatial language should ideally be evaluated comparatively to other existing approaches. I understand that the NER classes considered by the authors, useful for the type of study that they present, go beyond the classes that are considered in previous work. Still, it would be nice to see how the model compares to other approaches in terms of its ability to detect place names. Similarly, the method for "resolving spatial language" should also be compared against other alternatives (e.g., against existing open-source systems such as Mordecai).

**Reasons To Accept:**

* The paper describes creative/original work (interdisciplinary, although also on a very a niche topic that will not be of interest to all conference attendees), presenting an interesting analysis on the interplay between language, text and place.

* The paper is clear and well-written, properly contextualizing the problem and the contributions

* The authors plan to release the training data and models on Github, and detailed information about the annotation process and NER results is given in appendix.

**Reasons To Reject:**

* The descriptions associated to the methods for "resolving spatial language" are somewhat confusing. This part of the proposed approach was also not quantitatively evaluated, and it may be the case that this component is introducing many errors.

* Some of the choices regarding the methods for "detecting spatial language" should be better justified in the paper.

**Reproducibility:**

4: Could mostly reproduce the results, but there may be some variation because of sample variance or minor variations in their interpretation of the protocol or method.

**Reviewer Confidence:**

4: Quite sure. I tried to check the important points carefully. It's unlikely, though conceivable, that I missed something that should affect my ratings.

**Typos Grammar Style And Presentation Improvements:**

Section 4.2 is missing the F1 score the the DistilBERT-CRF model, and also the reference to the appendix with a full comparison of models (i.e., it mentions Appendix XX instead of Appendix A).

As mentioned before, Section 4.4 is somewhat difficult to follow, and perhaps it should be re-written.

---

> ### Author Rebuttal · Authors · 2023-08-28
>
> The authors would like to thank all three reviewers for their thoughtful, constructive, and knowledgeable feedback. We appreciate the opportunity to offer our responses here. Below, we respond to each reviewer’s “Reasons to reject” and “Questions for the authors” sections. For the sake of brevity we do not respond to “Reasons to accept,” “Missing references,” and “Typos” feedback; reviewers can assume that we will incorporate all suggested references and correct all typos in the final draft. We reproduce each reviewer comment in quotation marks below.
>
> Reasons to reject
>
> “The descriptions associated to the methods for "resolving spatial language" are somewhat confusing. This part of the proposed approach was also not quantitatively evaluated, and it may be the case that this component is introducing many errors.”
>
> We agree with this feedback and will devote more space to the “resolving” methods description, including a performance comparison of alternative approaches in an additional appendix. We already used internally a list of neighborhood synonyms (e.g. “Spanish Harlem” and “El Barrio” for East Harlem, or common misspellings like “Bushwhick”) to validate our model. We will motivate our choice of synonyms and include the list along with results for our chosen methods and some common alternatives for spatial outlier detection (e.g. Robust Covariance, SVM, Local Outlier Factor) in this appendix.
>
> “Some of the choices regarding the methods for "detecting spatial language" should be better justified in the paper.”
>
> The final draft will include more pointed justifications for our methodological choices, grounded in the appendix outlined above as well as relevant literature
>
> Questions for the authors
>
> “A - Section 4.1 should ideally report on the agreement between the annotators of the NER data, even if only leveraging on a small sub-set of the annotated data.”
>
> This is a fair point. The two main coders spent several meetings and considerable time together developing the coding scheme and protocols, including frequent discussions of edge cases during the coding process. However, we agree that an intercoder reliability test would be appropriate and will do so on a small sample as suggested.
>
> “B - Section 4.2 mentions that the DistilBERT-CRF model performed best in the NER experiments, but the section does not present the actual F1 metric (i.e., in states "an F1-score of XX"). The authors should also explain why they haven't considered using a larger BERT model for producing the representations, or why they haven't considered fine-tuning the DistilBERT model directly for the NER task (e.g., the "linear classifier" approach seems to correspond to the training of a linear layer on top of the BERT representations for each token, and this setting could be easily extended to consider also the training of the DistilBERT model itself). Ideally, additional technical details about the training and inner-workings of the model(s) should also be given. For instance, what were the hyper-parameters considered for model training (this is only shown in Appendix A, but it is somewhat strange that different values for parameters such as the batch size were used for the different models)? How was span recognition modeled as a token-classification problem in terms of tags associated to the tokens (e.g., did the authors use a B-I-O tagging scheme, or some other approach)? How was the BERT tokenization scheme, based on sub-words, combined with the tokenization scheme used for creating the annotated data?”
>
> We thank the reviewer for these questions, which are all very relevant. The F1-score was omitted from the paper due to human error and will be added in the final version. We will also run all our experiments with BERT and/or RoBERTa. We will add details about hyperparameters and run all models across similar batch sizes. For tagging, we used the BIO/IOB scheme. Subwords retained the tag of the actual word for model training. Predictions were then made by first tokenizing words using the BERT tokenization scheme, with subwords merged after prediction. We will add details on this to the appendix and a mention of it in the main text.
>
> “C - On what regards Section 4.4, the authors mention the use of a Kernel Density Estimation procedure to filter out locations more than three standard deviations from the mean of their coordinates. This procedure should be presented in more detail (e.g., what KDE parameters, such as the kernel function or the kernel bandwidth, were considered? What do the authors mean with the expression "mean of their coordinates"?), and the authors should also justify this choice against some other procedure to remove outliers. In connection to this, the authors also mention the use of a convex hull to delineate the region formed by the coordinates, but a more accurate procedure could have been used instead (e.g., threshold the results of the KDE function, to keep the high density region, or using a different computational geometry approach such as alpha-shapes, which is not restricted to the formation of convex polygons).”
>
> Thank you for these thoughtful methodological comments and suggestions. We will devote more discussion—depending on space, perhaps in an appendix—to describing and justifying the KDE/outlier removal process, up to and including a systematic comparison with other methods. The use of alpha-shapes is a good idea and we will try this approach. However, this method might result in regions that do not correspond to human intuitions of neighborhood shape. We will also experiment with using the high-density KDE region, per the reviewer's recommendations. Finally, we will replace the “mean of the coordinates” with the centroid of the region.
>
> “D - Also on what regards Section 4.4, which discusses methods for resolving spatial language, the authors mention the use of a "canonical list of toponyms." This should be further detailed in the paper. Does the canonical list correspond to a sub-set of geonames entries? The entire contents of Section 4.4 are actually somewhat confusing (e.g., it is also not clear why and how different string similarity metrics are used), and perhaps this section should be re-written.”
>
> We note the reviewer’s confusion and will try to improve the clarity of the section without sacrificing methodological detail. The “canonical list of toponyms” refers to a widely used shapefile for New York neighborhoods. We will add references to other papers that use this same list to clarify its status in research on neighborhood-level phenomena in New York.
>
> “E - The models used for detecting spatial language should ideally be evaluated comparatively to other existing approaches. I understand that the NER classes considered by the authors, useful for the type of study that they present, go beyond the classes that are considered in previous work. Still, it would be nice to see how the model compares to other approaches in terms of its ability to detect place names. Similarly, the method for "resolving spatial language" should also be compared against other alternatives (e.g., against existing open-source systems such as Mordecai).”
>
> This is a good idea and we thank the reviewer for the suggestion. We will our toponym tags and compare them with results from Mordecai as well as the Stanford NER parser “LOCATION” tags. Evaluating the toponymy resolution step is challenging because we don’t have extensive gold-standard labels. However, we will perform a limited evaluation using the validation set described in our response above.

---

### Meta-Review · Area_Chair_43ne · 2023-09-08

**Recommendation:** 5

**Metareview:**

In this paper, the authors use a named entity recognition model trained on a novel annotated dataset to analyze how property renters in New York City describe the locations of their listings on Airbnb. The authors connect this linguistic study with a spatial analysis of gentrification in New York. They also contextualize their findings using theories of “critical toponymy” from social science.

Overall, the review process converged to a consensus on this paper. The paper gained high soundness and excitement scores from all reviewers. Reviewers also offered very positive qualitative feedback.

Positives which emerged from the review process:
- All reviewers assigned a score of 4 for soundness and a 4 for excitement. One praised the paper for being “clear and well-written,” and “properly contextualizing” a computational analysis. Another noted that “the paper is well-written and I really enjoyed reading it.” A third described the paper as a “great example of high quality computational social science.”

Negatives which emerged from the review process:
- While the review process did surface some issues surrounding soundness in the original draft, these were addressed during the rebuttal period. Reviewer QGQw asked for missing details about KDE parameters, the NER tagging scheme, the outlier detection procedures and the F1 score for a specific model. Reviewer AMJs also asked for missing information about the correlation statistic. However, the authors provided detailed answers to these questions, and both reviewer QGQw and reviewer AMJs replied to the rebuttal to say that their concerns had been addressed. Each mentioned they still had a positive assessment of the paper after reading the rebuttal.

- Reviewer B1cM also noted that one limitation of this work is that it focuses on a single city (New York). The authors replied that this was due to practical considerations, and that their paper offers a proof-of-concept which might be applied to other cities in the future. They noted that annotated data for other cities could improve their analysis. B1cM read this rebuttal and did not change their already high score.

---

### Decision · Program_Chairs · 2023-10-07

**Decision:**

Accept-Main

**Comment:**

In this paper, the authors use a named entity recognition model trained on a novel annotated dataset to analyze how property renters in New York City describe the locations of their listings on Airbnb. The authors connect this linguistic study with a spatial analysis of gentrification in New York. They also contextualize their findings using theories of “critical toponymy” from social science.

Overall, the review process converged to a consensus on this paper. The paper gained high soundness and excitement scores from all reviewers. Reviewers also offered very positive qualitative feedback.

Positives which emerged from the review process:
- All reviewers assigned a score of 4 for soundness and a 4 for excitement. One praised the paper for being “clear and well-written,” and “properly contextualizing” a computational analysis. Another noted that “the paper is well-written and I really enjoyed reading it.” A third described the paper as a “great example of high quality computational social science.”

Negatives which emerged from the review process:
- While the review process did surface some issues surrounding soundness in the original draft, these were addressed during the rebuttal period. Reviewer QGQw asked for missing details about KDE parameters, the NER tagging scheme, the outlier detection procedures and the F1 score for a specific model. Reviewer AMJs also asked for missing information about the correlation statistic. However, the authors provided detailed answers to these questions, and both reviewer QGQw and reviewer AMJs replied to the rebuttal to say that their concerns had been addressed. Each mentioned they still had a positive assessment of the paper after reading the rebuttal.

- Reviewer B1cM also noted that one limitation of this work is that it focuses on a single city (New York). The authors replied that this was due to practical considerations, and that their paper offers a proof-of-concept which might be applied to other cities in the future. They noted that annotated data for other cities could improve their analysis. B1cM read this rebuttal and did not change their already high score.